# Lead Time and Prognostic Role of Serum CEA, CA19-9, IL-6, CRP, and YKL-40 after Adjuvant Chemotherapy in Colorectal Cancer

**DOI:** 10.3390/cancers13153892

**Published:** 2021-08-02

**Authors:** Kaisa Lehtomäki, Harri Mustonen, Pirkko-Liisa Kellokumpu-Lehtinen, Heikki Joensuu, Kethe Hermunen, Leena-Maija Soveri, Mogens Karsbøl Boisen, Christian Dehlendorff, Julia Sidenius Johansen, Caj Haglund, Pia Osterlund

**Affiliations:** 1Faculty of Medicine and Health Technology, Tampere University, Arvo Ylpön katu 34, 33520 Tampere, Finland; kaisa.lehtomaki@tuni.fi (K.L.); pirkko-liisa.kellokumpu-lehtinen@tuni.fi (P.-L.K.-L.); 2Department of Oncology: Tays Cancer Center, Tampere University Hospital, Teiskontie 35, 33520 Tampere, Finland; 3Department of Gastrointestinal Surgery, Helsinki University Hospital and University of Helsinki, P.O. Box 440, 00029 HUS Helsinki, Finland; harri.mustonen@helsinki.fi (H.M.); kethe.hermunen@hus.fi (K.H.); caj.haglund@helsinki.fi (C.H.); 4Research Programs Unit, Translational Cancer Medicine Program, University of Helsinki, P.O. Box 4, 00014 Helsinki, Finland; 5Department of Oncology, Helsinki University Hospital and University of Helsinki, Haartmaninkatu 4, 00290 Helsinki, Finland; heikki.joensuu@hus.fi; 6Hyvinkää Hospital and Hyvinkää Homecare, Sairaalankatu 1, 05850 Hyvinkää, Finland; leena-maija.soveri@keusote.fi; 7Department of Oncology, Herlev and Gentofte Hospital, Copenhagen University Hospital, Borgmester Ib Juuls vej 1, DK-2730 Herlev, Denmark; Mogens.Karsboel.Boisen@regionh.dk (M.K.B.); Julia.Sidenius.Johansen@regionh.dk (J.S.J.); 8Statistics and Data Analysis Danish Cancer Society Research Center, Danish Cancer Society, Strandboulevarden 49, DK-2100 Copenhagen, Denmark; cdchr395@hotmail.com; 9Department of Medicine, Herlev and Gentofte Hospital, Copenhagen University Hospital and University of Copenhagen, Borgmester Ib Juuls vej 1, DK-2730 Herlev, Denmark; 10Institute of Clinical Medicine, Faculty of Health and Medical Sciences, University of Copenhagen, Blegdamsvej 3, DK-2200 Copenhagen, Denmark; 11Department of Oncology-Pathology, Karolinska Institutet, Karolinska vägen A2:07, 171 64 Stockholm, Sweden; 12Tema Cancer-Bäckensektionen, Karolinska sjukhuset, Eugeniavägen 3, 171 64 Stockholm, Sweden

**Keywords:** colorectal cancer, prognostic biomarker, tumour marker, CEA, CA19-9, IL-6, CRP, YKL-40, post-adjuvant, lead time

## Abstract

**Simple Summary:**

Colorectal cancer is the third most common cancer worldwide. Recurrence risk after curative intent surgery combined with adjuvant chemotherapy is substantial. Unlike many other cancers, curative metastasectomy is possible upon recurrence, which raises the question of personalized surveillance strategies according to individual risk factors. We studied whether elevated biomarkers, such as gold standard CEA and experimental CA19-9, IL-6, CRP, and YKL-40 after adjuvant therapy, are associated with disease-free and/or overall survival, and whether the diagnostic time from the elevated biomarker to the diagnosis of metastases can be prolonged by combining these biomarkers. We show that elevated post-adjuvant CEA, IL-6, and CRP are associated with impaired survival and that elevated IL-6 finds recurrences in patients with normal CEA. Lead time is shorter with CEA than with experimental biomarkers. Our findings thus may impact the follow-up strategies after curative intent treatment aiming at finding operable relapses. These biomarkers are readily available and feasible in clinical practice.

**Abstract:**

In colorectal cancer (CRC), 20–50% of patients relapse after curative-intent surgery with or without adjuvant therapy. We investigated the lead times and prognostic value of post-adjuvant (8 months from randomisation to adjuvant treatment) serum CEA, CA19-9, IL-6, CRP, and YKL-40. We included 147 radically resected stage II–IV CRC treated with 24 weeks of adjuvant 5-fluorouracil-based chemotherapy in the phase III LIPSYT-study (ISRCTN98405441). All 147 were included in lead time analysis, but 12 relapsing during adjuvant therapy were excluded from post-adjuvant analysis. Elevated post-adjuvant CEA, IL-6, and CRP were associated with impaired disease-free survival (DFS) with hazard ratio (HR) 5.21 (95% confidence interval 2.32–11.69); 3.72 (1.99–6.95); 2.58 (1.18–5.61), respectively, and elevated IL-6 and CRP with impaired overall survival (OS) HR 3.06 (1.64–5.73); 3.41 (1.55–7.49), respectively. Elevated post-adjuvant IL-6 in CEA-normal patients identified a subgroup with impaired DFS. HR 3.12 (1.38–7.04) and OS, HR 3.20 (1.39–7.37). The lead times between the elevated biomarker and radiological relapse were 7.8 months for CEA and 10.0–53.1 months for CA19-9, IL-6, CRP, and YKL-40, and the lead time for the five combined was 27.3 months. Elevated post-adjuvant CEA, IL-6, and CRP were associated with impaired DFS. The lead time was shortest for CEA.

## 1. Introduction

Colorectal cancer (CRC) ranks third in incidence and second in mortality among all malignancies worldwide [1]. After a curatively aimed primary surgery, patients with a high risk of recurrence are referred for adjuvant chemotherapy [2] and followed for up to 5 years for signs of cancer recurrence using mainly carcinoembryonic antigen (CEA), colonoscopy, and radiological imaging [3,4]. CEA is a well-known plasma membrane-anchored glycoprotein that was described in 1965 [5] and remains the only circulating biomarker recommended for clinical use in patients with CRC [3,4]. Despite technical advances in imaging, no additional survival benefit was observed when concomitant computed tomography (CT) imaging and CEA were used in a prospective randomised follow-up study [6]. The prognostic value of both preoperative and postoperative CEA has been described in several retrospective and post hoc studies [7,8,9,10,11]. However, the best surveillance algorithm for patients resected for stage II and III CRC remains unknown.

The half-life of serum CEA is known to be approximately 7 days, and its levels should normalise within 4 to 6 weeks after macroscopically curative surgery. Sustained elevation of CEA may indicate residual disease [12]. Serum levels of CEA often increase prior to new cancer-related symptoms or identification of recurrence on imaging with a median lead time of 4.5–8 months [10]. Nevertheless, no difference was observed between the CRC patients with short lead times or those with longer lead times (over 3 months) and the rate of metastasectomy (20.1% versus 17.3%) or overall survival (OS) [13]. A meta-analysis of randomised trials investigating follow-up after resection of primary CRC showed that more intensive monitoring shortened the time to diagnosis of recurrence by a median of 10 (interquartile range 5–24) months [14] but did not, however, result in a statistically significant difference in all-cause mortality. Salvage surgery frequency was doubled by intensive follow-up according to a Cochrane review, but this intervention was still not sufficiently common to result in an overall survival improvement for the group as a whole [15].

The optimal timing of CEA determination after chemotherapy has been debated [16], and several studies have investigated the dynamics of CEA during and after chemotherapy in patients with metastatic CRC (mCRC) [16,17]. However, far less is known about the implications of elevated post-adjuvant chemotherapy biomarker levels in patients with localised CRC treated with a curative aim [18]. When a threshold of 5 µg/L is used for CEA, the pooled sensitivity for recurrence is 71%, and the specificity is 88% [8]. Unfortunately, approximately 20% of CRCs are CEA-negative [8]. Due to this insufficient sensitivity, additional methods for the early detection of CRC recurrence are urgently needed [8,19,20].

Carbohydrate antigen 19-9 (CA19-9) is another widely used serum biomarker in gastrointestinal malignancies. It is elevated in 14–67% of patients with CRC [21], but its prognostic value as single biomarker seems weak [8,21,22,23]. CA19-9 is often used in combination with CEA, even though the prognostic role of CA19-9 in patients with CRC is poorly elucidated [17,21,24]. 

It is well-established that inflammation influences the manifestation and progression of CRC [20]. Interleukin 6 (IL-6) is a cytokine produced during acute and chronic inflammation [25]. IL-6 affects many hallmarks of cancer, such as proliferation, cell growth, and inhibition of apoptosis, and can enable the tumour cells to become drug-resistant [25,26,27]. Elevated preoperative IL-6 has been shown to be a prognostic factor of impaired DFS and OS in CRC [28]. C-reactive protein (CRP), an acute-phase plasma protein, is generated in hepatocytes in response to inflammatory cytokines such as IL-1, TNF-α, and, in particular, IL-6 [25]. An increase in CRP concentration correlates with poor prognosis in both localised [29] and metastatic CRC [30]. YKL-40, also known as human cartilage glycoprotein-39 or chitinase-3-like protein 1, is another biomarker of inflammation and plays a role in the differentiation of macrophages, extracellular matrix remodelling, and organisation and migration of endothelial cells. YKL-40 is secreted by tumour cells, such as colon cancer cells, and by tumour-associated macrophages [31], and promotes cancer proliferation and inflammation [32]. Elevated YKL-40 expression in tumours and in the circulation is associated with poor prognosis in various tumour types, including CRC [32,33,34].

Approximately 20–30% of patients resected for stage II–III CRC will experience recurrence even after radical surgery and adjuvant therapy, and there is no method in clinical use to detect the residual disease in these patients. Thus, there is an unmet need for new post-adjuvant prognostic biomarkers [2,35]. Circulating tumour DNA (ctDNA)-based technologies have created a lot of enthusiasm in the detection of CRC [36,37] but may be methodologically challenging and are not yet widely used.

The primary aim of this study was to investigate the utility of the post-adjuvant levels of serum CEA and four other serum biomarkers, CA19-9, IL-6, CRP, and YKL-40, as prognostic biomarkers for disease-free survival (DFS) and OS in CRC patients treated with curative intent. A secondary aim was to determine the lead times between the first detection of an elevated biomarker level and diagnosis of disease recurrence.

## 2. Materials and Methods

### 2.1. Patients

The LIPSYT trial was an open-label, prospective, randomised single-institution study in patients with radically resected CRC (ISRCTN98405441). The patients accrued received adjuvant chemotherapy at the Department of Oncology of Helsinki University Hospital, Finland, between November 1997 and August 2001. The primary aim was to assess treatment tolerability in a two-by-two factorial design trial; the secondary aim was to study biomarkers. The patients were randomly allocated to receive either adjuvant chemotherapy consisting of 5-FU and leucovorin (LV) administered as a bolus injection (the Mayo regimen) or continuous 5-FU infusion (simplified de Gramont regimen) [38].

The LIPSYT trial included 150 patients, of whom 3 never started treatment; thus, 147 randomised patients were included in the lead time analysis and a time-dependent Cox regression analysis. Twelve (8%) patients relapsed during adjuvant treatment, i.e., within the first 8 months from the date of randomisation and were excluded from all analyses apart from the lead time and time dependent analyses. Thus, 135 patients were included in the post-adjuvant analysis. Inclusion criteria were age 18 to 75 years, histologically confirmed, radically resected stage II–IV CRC (stage IV included 18 patients with radical metastasectomy of mostly liver metastases), the World Health Organisation performance status 0–2, and adequate bone marrow, kidney, and liver function. Exclusion criteria included history of invasive cancer other than CRC; metabolic, neurological, or psychiatric illness incompatible with chemotherapy; serious thromboembolic event currently under treatment; and pregnancy, lactation, or absence of adequate contraception in fertile patients. The study was conducted according to the guidelines of the Declaration of Helsinki. The protocol was approved by the institutional review board at Helsinki University Hospital (5 November 1997), and all study participants gave their signed informed consent. 

### 2.2. Assessment of Biomarkers

Serum biomarkers CEA and CA19-9 were analysed prospectively as part of the clinical routine, and IL-6, CRP, and YKL-40 were measured post hoc in samples collected postoperatively (baseline) before adjuvant chemotherapy and at 4 months, 8 months (approximately 2 months after completion of adjuvant chemotherapy and 10 months from surgery), 1 year, 2 years, 3 years, 5 years, and 10 years from the date of randomisation. The median time to the first postoperative sampling was 48 days (range 19–124 days). Time to treatment initiation was more than 8 weeks from the date of surgery in 40 patients (27%) due to a referral delay in 33 patients, and 7 patients were unfit to start adjuvant chemotherapy within 8 weeks.

CRP, CEA, and CA19-9 were determined by the accredited methodology as follows. CRP: immunoturbidimetric method at HUSlab laboratories, Helsinki University Hospital; CEA and CA19-9: immunoenzymatic assay, Bayer Immuno 1 (CEA: October 1998 to October 2005; CA19-9: January 1998 to January 2006), or immunochemiluminometric assay, Abbott Architect (CEA: October 2005→ and CA19-9: January 2006→). All measurements were performed by technicians blinded to study endpoints. 

Blood samples for YKL-40 and IL-6 were collected in gel tubes and centrifuged within 2 h; serum was stored at −20 °C until analysis. YKL-40 and IL-6 were determined in duplicate with commercially available enzyme-linked immunosorbent assays (ELISAs)—YKL-40: MicroVue YKL-40 ELISA (Catalog #8020), Quidel Corporation, San Diego, CA, USA and IL-6: Quantikine HS600B, R&D Systems, Abingdon, OX, UK—according to manufacturer’s instructions. For YKL-40, the detection limit was 20 ng/mL, and intra- and inter-assay coefficients of variation (CVs) were of <5% and <6%, respectively. For IL-6, detection limit was 0.01 pg/mL, and intra- and inter-assay CVs were of ≤8% and ≤11%, respectively.

An age-corrected percentile for YKL-40 was calculated according to the formula: percentile = 100/(1 + (YKL-40^−3^) * (1.062 ^ age) * 5000). Cut-off values were according to Hermunen et al. [7]; age-corrected YKL-40 level as the 70.7th percentile of normal controls YKL-40; 4.5 pg/mL for IL-6; 10 mg/L for CRP; 5 µg/L for CEA; and 26 kU/L for CA19-9. Continuous log_2_ transformed biomarker values were used in time varying analyses, otherwise elevated over cut-off versus normal values were used.

### 2.3. Statistics 

Clinicopathological parameters and tumour-marker values are presented as frequencies or medians with range for nonparametric distributions. Overall survival and DFS were estimated with the Kaplan–Meier estimator overall and according to subgroups. DFS was defined as the time period from the date of randomisation to the date of recurrence or death from any cause with censoring patients alive without recurrence on the last date of follow-up. OS was defined as the time period from the date of randomisation to the date of death from any cause censoring patients alive on the last date of follow-up. Unadjusted and adjusted hazard ratios (HRs) and 95% confidence intervals (CI) were estimated with the Cox regression proportional hazard model. Adjustments were made for age, sex, inflammatory disease, and TNM stage in adjusted analyses. Inflammatory diseases possibly affecting YKL-40 concentrations were rheumatoid arthritis, iritis, psoriasis, non-active ulcerative colitis, coeliac disease, and thyroiditis in the history of 14 patients in this study. The Cox regression assumption of constant hazard ratios over time (proportional hazards) was assessed with the Schoenfeld residuals plotted over time and testing for a trend. There were slight indications of nonproportional hazards with CEA and CA19-9, and therefore, in the secondary analyses (the modified model), the time axis was split into 2 using the survSplit function in R for delayed entry into the model, giving separate estimates for the two time periods. To analyse with time varying biomarker levels with the Cox regression analysis, time-dependent data sets were built with the tmerge function in R to create multiple start/stop intervals per subject.

The lead time was defined as the time from elevated biomarker value (above cut off) to diagnosis of relapse, first separately for all five biomarkers and then combined as any of the five biomarkers elevated. The median for lead time was estimated from these intervals censored by data from Weibull regression. Time-dependent receiver operating characteristic (ROC at 1 and 7 years after the start of follow-up—i.e., from randomisation) curves for biomarkers were constructed, relapse as event and other non-CRC deaths as competing events. The area under the curve (AUC) values were determined with the time-ROC package in R (https://cran.r-project.org/web/packages/timeROC/ v0.4 by Paul Blanche, 2019) (accessed on 28 April 2021).

The statistical significance level was set at 5%; all tests are 2-sided. Statistical analyses were done with SPSS version 26.0 (IBM SPSS Statistics, version 22.0 for Mac; SPSS, Inc., Chicago, IL, USA), R version 3.6.1 (Foundation for Statistical Computing, Vienna, Austria), and STATA/MP version 15.1 (StataCorp LLC, College Station, TX, USA).

## 3. Results

### 3.1. Baseline Characteristics and Biomarker Levels

The total study cohort that was used in the lead time and time-dependent Cox analyses encompassed 147 patients (Table 1). Their median age was 60 years. Most presented with locoregional disease (88% had stage II or III), but 18 (12%) had undergone radical metastasectomy, mostly liver resections. Twelve patients had a recurrence during adjuvant treatment and were excluded from the post-adjuvant biomarker analysis. Median follow-up time for living patients was 11.9 (range, 8.9–12.7) years.

The 5- and 10-year DFS rates were 54% and 50%, and the 5- and 10-year OS rates were 69% and 55%, respectively. Relapse was detected in 65 patients (44%), with no new disease relapses after 6.3 years. Twenty-six (40%) of the patients with a relapse had a metastasectomy, of which seven (27%) never experienced a second relapse. The remaining relapsed patients received palliative chemotherapy, of whom two had a long-lasting complete response. The cause of death was mCRC in 84% (58/70), cardiovascular disease in 10% (7/70), secondary cancer in 3% (2/70), and other causes in 4% (3/70).

The CEA and CA19-9 levels were elevated in 11–14% of patients postoperatively and in 10–12% post-adjuvant, and the inflammatory markers IL-6, CRP, and YKL-40 were elevated in 12–44% postoperatively and 8–49% post-adjuvant (Table 2). These elevated post-adjuvant proportions were 11–57% in patients with relapse during follow-up. The medians and ranges are summarised in Table 2, and Appendix A shows changes, medians, and percentiles of all five biomarkers during surveillance.

### 3.2. Association between Elevated Biomarkers in the Post-Adjuvant Setting, i.e., at 8 Months from Randomisation, and DFS or OS

In an analysis adjusted for baseline characteristics (age, sex, inflammatory disease, TNM stage, and chemotherapy regimen), elevated CEA, IL-6, and CRP were associated with impaired DFS (Figure 1A,C,D). Due to a small sample size, no statistically significant association was noted between elevated CA19-9 or YKL-40 and DFS (Figure 1B,E).

Elevated IL-6 and CRP were associated with impaired OS (Figure 2C,D). Due to the small number of patients, no statistically significant association with OS was found for elevated CEA, CA19-9, or YKL-40 (Figure 2A,B,E). Second, metastasectomy with curative intent affected OS (23 out of 54 relapsed patients), seen especially in plateaus in OS estimates for prospectively measured CEA and CA19-9 (Figure 2).

Due to indications of nonproportional hazards in Schoenfield residuals for elevated CEA and CA19-9 in the DFS analysis (Appendix A), we further investigated these biomarkers in a modified model, where the time axis was split at 12 months: for the first year after primary surgery and for the time after that. In this modified analysis, adjusted for the abovementioned baseline characteristics, elevated CEA was associated with impaired DFS for the first year after primary surgery but not later during follow-up. Similar to CEA, elevated CA19-9 was associated with impaired DFS for the first year after primary surgery, but not later (Appendix A).

### 3.3. Association between Post-Adjuvant, i.e., at 8 Months after Randomisation, IL-6 and DFS or OS within TNM Stages

No statistically significant association was noted between elevated post-adjuvant IL-6 and DFS or OS in the small subgroup with stage II disease (Appendix A). Associations between elevated IL-6 and impaired DFS or OS were observed in patients with stage III–IV disease (Appendix A).

### 3.4. Mutually Adjusted Multivariable Model of DFS and OS for All Biomarkers Measured in the Post-Adjuvant Setting

In the mutually adjusted multivariable analysis of all five biomarkers combined, post-adjuvant elevated CEA and IL-6 were associated with impaired DFS, and elevated levels of IL-6 and CRP were associated with impaired OS (Table 3).

In the mutually adjusted modified model of all five biomarkers combined, where the time axis was split at 12 months, elevated CEA and CA19-9 were associated with impaired DFS for the first year after primary surgery, but no significant associations were seen later (Appendix A).

### 3.5. Post-Adjuvant Normal CEA Combined with Elevated CA19-9, CRP, IL-6, and YKL-40

In patients with normal CEA (<5 µg/L), elevated IL-6 was associated with impaired DFS and OS (Figure 3). Elevated CA19-9, CRP, or YKL-40 showed no statistically significant associations with DFS or OS in patients with normal CEA levels (Appendix A).

### 3.6. Diagnostic Accuracy for Postoperative Serum Biomarker Levels, i.e., before Adjuvant Treatment, and Diagnosis of Recurrence

We estimated the diagnostic accuracy for relapse at 1 and 7 years, with non-CRC deaths as a competing event, in relation to the baseline biomarker level. Appendix A shows the time-dependent area under the curve (AUC) values for this analysis for all five biomarkers. The AUC of the receiver operating characteristic (ROC) curve at 1 year was better for CEA (0.82) than for the other biomarkers (AUC = 0.60–0.69). At 7 years, AUCs were higher for IL-6 and YKL-40 (AUC = 0.65–0.67) than for CEA, CA19-9, and CRP (AUC = 0.55–0.64).

### 3.7. Lead Times, i.e., Interval between Elevated Serum Biomarker and Diagnosis of Recurrence, during Surveillance

The lead time, i.e., from first timepoint of elevated serum CEA (measured at predefined intervals from postoperative to 10 years) to the radiological or clinical diagnosis of a local relapse or distant metastasis, was 7.8 months (Table 4). The lead time was 10.0 months for CA19-9, 21.8 months for IL-6, 10.2 months for CRP, and 53.1 months for YKL-40 (Table 4). The lead time from first timepoint of elevation of any (one or more) of the five biomarkers elevated to relapse was 27.3 months (Table 4).

### 3.8. Time-Varying Serum Biomarkers during Surveillance in the Cox Regression Model and DFS

The time-varying biomarker Cox-regression models were adjusted for age, sex, inflammatory disease, TNM stage, and chemotherapy regimen. A 2-fold increase during surveillance in CEA values starting from the postoperative values was linked to a 2.1-fold higher risk of relapse or death during follow-up. A 2-fold increase in CA19-9, IL-6, CRP, and YKL-40 was linked to a 1.4-, 1.5-, 1.6-, and 1.3-fold increase in the risk of relapse or death, respectively (Table 5, Appendix A).

## 4. Discussion

The goal of our study was to discover prognostic biomarkers of survival after adjuvant chemotherapy in patients with CRC and to determine the lead times between biomarker elevation and diagnosis of clinically detectable recurrence. Our results indicate that elevated post-adjuvant CEA, IL-6, and CRP levels are independent predictors of survival. CEA had the shortest lead time, and by combining one or more biomarkers, the lead time was extended nearly threefold. 

Several studies have investigated the prognostic importance of CEA and other biomarkers in the pre- and/or postoperative setting of radically resected CRC [7,8,9,10,11,20,39,40]. Interestingly, very few biomarker studies have investigated the post-adjuvant time point [17,18], which is one of the most important in ctDNA research [36,41]. In a prospective study, elevated post-adjuvant microRNAs, such as miR-17, miR-21, or miR-92, were associated with early recurrence of CRC [18]. In a retrospective analysis, Sakamoto et al. showed that post-chemotherapeutic levels of CEA and CA19-9 were prognostic in patients treated with curative intent metastasectomy for CRC liver metastases [17]. High post-adjuvant CA19-9 was an independent prognostic factor for impaired recurrence-free survival (RFS) in a retrospective study of pancreatic ductal adenocarcinoma [42]. Some recurrences occur soon after adjuvant therapy, which might be a sign of resistance to the given therapy, and are demonstrated by elevated markers after adjuvant therapy [2]. This time point is also of utmost importance, as shown in studies of ctNDA, where post-adjuvant elevated ctDNA was linked to a very high recurrence rate and where the clearance of ctDNA with adjuvant therapy was connected with a clearly lower relapse risk [41].

The studied biomarkers are known to correlate with each other. CEA has been shown to induce the release of serum IL-6 [43,44], and IL-6 in turn induces CRP and YKL-40 secretion [25,29]. High preoperative serum IL-6 levels are thus associated with elevated CEA and CRP levels, with negative prognostic factors such as colonic obstruction and T4 extension, and also with impaired DFS and OS, especially in stage II disease [28]. In our study, elevated levels of IL-6 and CRP are associated with impaired OS, while elevated levels of CEA are not. Since CEA is the only biomarker in clinical use and its elevation leads to imaging interventions and possibly metastasectomy, the lacking association with OS is understandable. CA19-9 was also measured prospectively, and elevated values could have led to metastasectomies, thus possibly interfering with OS.

In a TNM-adjusted combined analysis, we show that post-adjuvant, elevated levels of CEA and IL-6 were prognostic indicators of impaired DFS, whereas CA19-9, CRP, and YKL-40 did not add prognostic value. Many researchers have adopted a strategy of combining multiple biomarkers to better identify patients with a high risk of CRC recurrence. Some have used the same biomarkers as in this study [7,45], while some have used additional biomarkers, such as D-dimer, Glasgow prognostic score, CRP, CRP/albumin ratio, and neutrophil-lymphocyte ratio [20], and plasma miRNA levels combined with CEA and CA19-9 [46]. Elevation of biomarkers, e.g., CEA, CA19-9, YKL-40, IL-6, and CRP, postoperatively or in conjunction with metastasectomy for mCRC has been associated with impaired OS [7,45]. In patients with localised CRC with normal CEA, elevated YKL-40 or CRP identified patients with a high risk of relapse [7]. These results illustrate the additive prognostic value of combining inflammatory biomarkers with CEA. Another reasonable approach is augmenting the sensitivity of CEA by combining it with ctDNA [41]. 

CEA has a lead time of 7.8 months in our study, which is concordant with earlier observations of lead time ranging from 4.5 to 8 months when the detection limit of 5 µg/L was used for CEA [10,47]. Park et al. reported a shorter lead time of 2.5 months but used a higher cut-off (7 µg/L) [39]. Unlike patients in other studies, all the patients in our study received adjuvant chemotherapy, which may have prolonged the lead time. In their prospective study, Barillari et al. showed that the lead time of CEA, tissue plasminogen activator (TPA), and/or CA19-9 was only 2 months in patients with liver metastases and 4 months in patients with extrahepatic metastases [48]. A shorter lead time for liver metastases has also been previously described [10]. Based on the modified model, it seems as though CEA measurement would predict a recurrence only in the near future.

Several studies have investigated whether intensified follow-up or addition of other surveillance methods along with CEA could offer a survival benefit by advancing the diagnosis of metastatic disease and thus enabling metastasectomy [14,15]. To date, no survival benefit has been noted in retrospective studies [49,50], prospective studies [13], meta-analyses [14,15], or randomised trials [47,51]. In our study, prospective CEA and CA19-9 measurements were combined with yearly radiology impacting biomarker results. In line with the abovementioned studies, our results show that other methods are needed along with CEA, as it has a short lead time and decreasing AUC after one year. We found that all the other markers investigated have a longer lead time than CEA, with maintained or improved AUC for IL-6, CRP, and YKL-40 at 1 versus 7 years, whereas the diagnostic accuracy of CA19-9 decreases similarly to CEA. The diagnostic interval may be extended to 27 months by combining the five biomarkers, but the small sample size limits the generalisation of this finding. When we investigated the risk increase in a time-dependent Cox regression model, a twofold increase in the CEA level indicated a 2.1-fold risk of relapse or death, and thus, CEA was clearly the strongest marker compared with CA19-9, IL-6, CRP, and YKL-40 with HRs of 1.3–1.6.

In the modified models, analogous to their shorter lead times (7.8–10.0 months), elevated CEA or CA19-9 is able to identify early recurrences (during the first year) but not later ones, which was also reflected in the OS results. Elevated IL-6 or CRP, with longer lead times of 21.8 and 10.2, respectively, seemed to reveal recurrences that occurred later, which is also depicted in the ROC curves. These results support the concept of using multiple biomarkers, as they capture different patterns and timings of recurrence, as well as different tumour characteristics such as differentiation, which at least affects the CEA levels [10]. Our findings suggest that CA19-9, IL-6, and CRP could provide additive information along with CEA in the post-adjuvant setting.

We showed that elevated post-adjuvant IL-6 levels in CEA-negative patients identified a subgroup of patients with a higher risk for recurrence and death. To our knowledge, this is the first study to show the prognostic significance of IL-6 in the post-adjuvant setting. Since IL-6 is known to participate in resistance to anticancer treatment [25,26,27], this finding is of special interest after adjuvant therapy.

Our finding that IL-6 predicts impaired DFS and OS in patients with more advanced stages of disease is in agreement with previously published findings [27,28]. It has been suggested that IL-6 could be a biomarker of a more aggressive tumour biology and that higher IL-6 concentrations are also correlated with impaired survival in several other cancer types as well [27].

Some evidence indicates that high pretreatment levels of IL-6 alone in localised CRC [28] and in combination with CRP or YKL-40 in mCRC [45,52] can be indicative of impaired prognosis. In contrast, some studies demonstrated no association between IL-6 and OS in localised [53] or metastatic [54] CRC. Due to contradictory results, IL-6 and its correlation with prognosis remain elusive [27].

The strengths of our study are the mature data with long follow-up times and no patients lost to follow-up. Additionally, clinical and radiologic examinations were performed on a yearly basis in all of our patients, and asymptomatic recurrences were found in patients with ever-normal CEA and CA19-9. We used an easily accessible multiple protein biomarker panel. However, the study was conducted in the pre-oxaliplatin era, which weakens the generalisability of our results. The small sample size also resulted in low power for all the statistical analyses, especially for the subgroup analyses.

Thus, there is an urgent need to add prognostic value to the current follow-up standards, which can be achieved with the combination of validated biomarkers [35]. Unlike many other GI cancers, curative metastasectomy is possible in CRC, which raises the question of personalised surveillance strategies according to individual risk factors [6,19,55]. Our results outline the importance of the post-adjuvant time point as a landmark in planning the follow-up.

## 5. Conclusions

Our study showed that patients with elevated levels of CEA, CA19-9, CRP, or IL-6 after adjuvant therapy are at higher risk of recurrence or death compared to those with normal levels. We also showed that elevated IL-6 remains indicative of a higher risk for recurrence or death after adjuvant therapy in patients with normal CEA levels. Since CEA is currently the only biomarker in clinical use for CRC surveillance, we suggest the addition of IL-6 to the post-adjuvant surveillance programme to better serve the CEA-negative patients in need of intensified surveillance. We found that all the other markers investigated had a longer lead time than CEA. Therefore, the diagnostic interval can be extended by combining biomarkers.

## Figures and Tables

**Figure 1 cancers-13-03892-f001:**
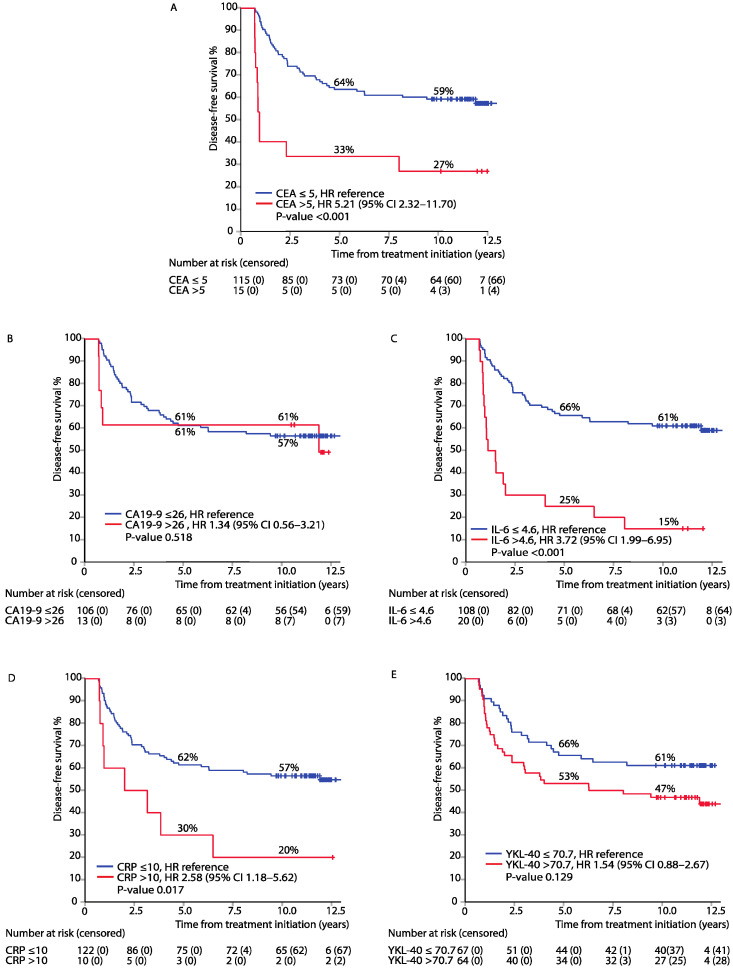
Disease-free survival (DFS) after adjuvant therapy in patients with elevated versus normal biomarker levels: CEA (**A**), CA19-9 (**B**), IL-6 (**C**), CRP (**D**), and YKL-40 (**E**); adjusted hazard ratios (HR) and 95% confidence intervals (95% CI).

**Figure 2 cancers-13-03892-f002:**
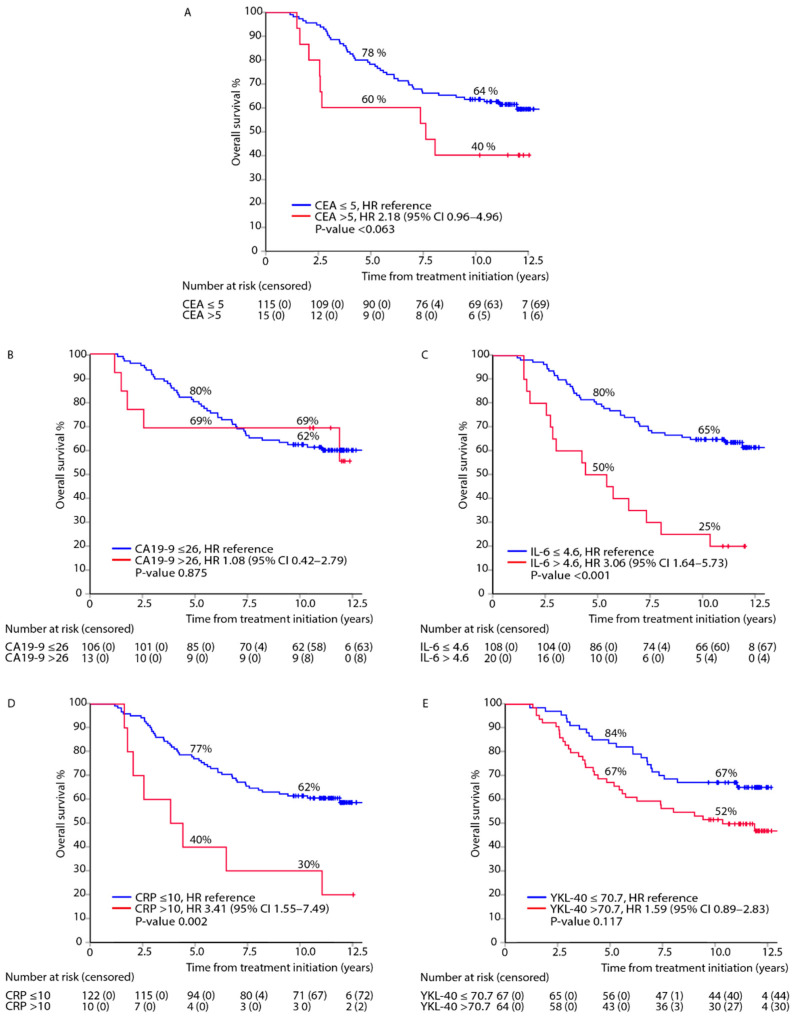
Overall survival (OS) after adjuvant therapy in patients with elevated versus normal biomarker levels: CEA (**A**), CA19-9 (**B**), IL-6 (**C**), CRP (**D**), and YKL-40 (**E**); adjusted hazard ratios (HR) and 95% confidence intervals (95% CI).

**Figure 3 cancers-13-03892-f003:**
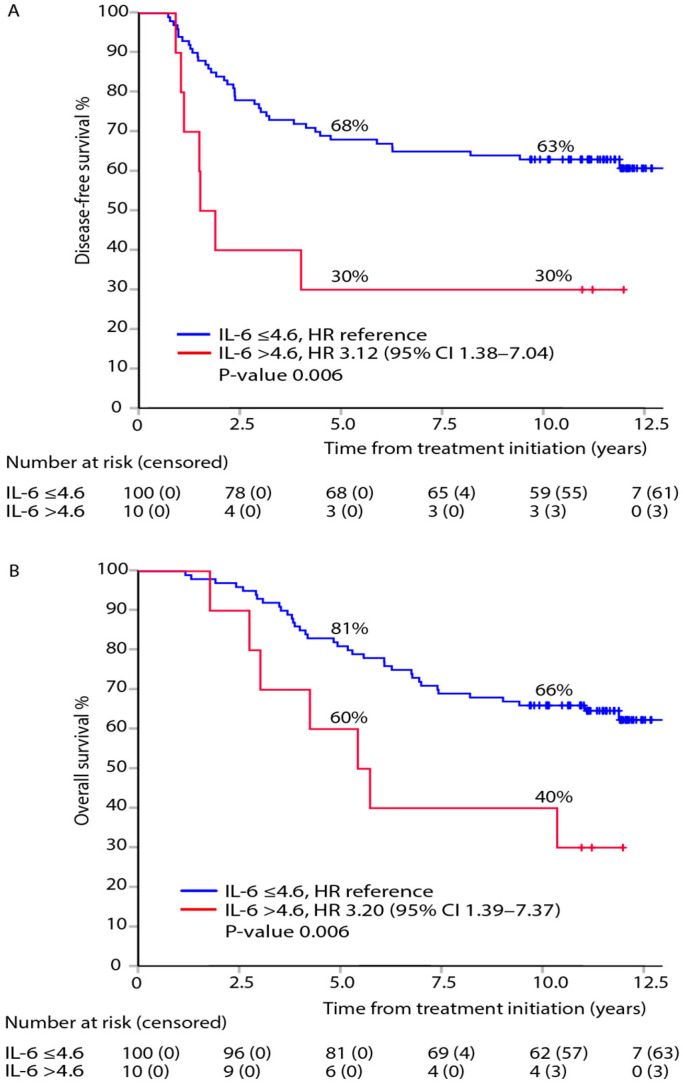
Association between elevated post-adjuvant IL-6 and outcomes: DFS (**A**) and OS (**B**), in the subgroup of patients with normal post-adjuvant CEA, with adjusted hazard ratios (HR) and 95% confidence intervals (95% CI).

**Table 1 cancers-13-03892-t001:** Patient demographics.

		All Patients		No Relapse at 8 Months	
		*n* = 147	%	*n* = 135	%
Age	Median years	60.3		60.4	
	Range	(31.1–75.9)		(31.1–75.9)	
	<70	131	89%	119	88%
	≥70	16	11%	16	12%
Sex	Male	75	51%	71	53%
	Female	72	49%	64	47%
Inflammatory disease *	No	133	91%	121	90%
	Yes	14	10%	14	10%
Chemotherapy regimen	5FU + LV bolus inj.	75	51%	70	52%
	5FU + LV continuous inf.	72	49%	65	48%
Primary location	Right colon	41	28%	39	29%
	Left colon	46	31%	42	31%
	Rectal	60	41%	54	40%
Radiotherapy for rectal primary	No	9	15%	8	15%
	Preoperative	8	13%	8	15%
	Postoperative	43	72%	38	70%
TNM stage	IIA-B	38	26%	38	28%
	IIIA-C	91	62%	85	63%
	IV	18	12%	12	9%
Relapse site	No relapse	81	55%	81	60%
	Only local	13	9%	10	7%
	Distant metastases	53	36%	44	33%

* Inflammatory diseases adjusted for: autoimmune diseases such as rheumatoid arthritis, iritis, psoriasis, ulcerative colitis, coeliac disease, and thyroiditis. 5FU + LV = 5-fluorouracil and leucovorin with randomisation to bolus injection (Mayo regimen) or continuous infused (de Gramont regimen).

**Table 2 cancers-13-03892-t002:** Postoperative and post-adjuvant biomarker levels. Number of patients with relapse, without relapse, metastasectomy, and non-CRC-related death during follow-up.

		Postoperative	Post-Adjuvant				
		0 Months	8 Months from Randomisation	No Relapse	Relapse	Metastasectomy	Non-CRC Death
		*n* = 147	*n* = 135	*n* = 81	*n* = 54	*n* = 23	*n* = 12
CEA	*n*	132	130	79	51	22	11
	Median (range) (µg/L)	1.9 (<1–305)	2.5 (<1–111)	2.3 (<1–7)	2.6 (<1–111)	2.3 (<1–15)	2.1 (<1–8)
	Elevated (>5 μg/L), *n* (%)	18 (14)	15 (12)	5 (6)	10 (20)	3 (14)	2 (18)
CA19-9	*n*	111	119	74	45	22	10
	Median (range) (kU/L)	6.0 (<5–2003)	6.0 (<5–902)	6.5 (<5–108)	<5 (<5–902)	<5 (<5–27)	7.0 (<5–27)
	Elevated (>26 kU/L), *n* (%)	12 (11)	13 (10)	8 (11)	5 (11)	1 (5)	1 (10)
IL-6	*n*	143	128	76	52	22	11
	Median (range) (pg/mL)	2.3 (0.4–36)	1.9 (0.2–25)	1.6 (0.2–10)	2.4 (0.7–25)	1.8 (0.7–25)	2.2 (1–10)
	Elevated (>4.5 pg/mL), *n* (%)	24 (17)	20 (16)	5 (7)	15 (29)	4 (18)	4 (36)
CRP	*n*	146	132	79	53	22	11
	Median (range) (mg/L)	<5 (<5–174)	<5 (<5–175)	<5 (<5–15)	<5 (<5–175)	<5 (<5–14)	<5 (4–15)
	Elevated (>10 mg/L), *n* (%)	17 (12)	10 (8)	4 (5)	6 (11)	2 (9)	2 (18)
YKL-40	*n*	144	131	78	53	23	11
	Median (range) (ng/mL)	64.5 (20–1524)	68.0 (20–1140)	63.5 (20–230)	89 (20–1140)	66 (20–175)	84 (34–203)
	Elevated (>70.7), *n* (%)	63 (44)	64 (49)	34 (44)	30 (57)	10 (43)	7 (64)

**Table 3 cancers-13-03892-t003:** Mutually adjusted multivariable Cox model for DFS and OS of elevated versus normal CEA, CA19-9, IL-6, CRP, and YKL-40 combined.

	DFS				OS	
	HR	95% CI	*p*-Value	HR	95% CI	*p*-Value
Adjusted for TNM Stage *n* = 112						
CEA elevated vs. normal	2.57	1.03–6.39	0.043	0.99	0.36–2.70	0.980
CA19-9 elevated vs. normal	1.17	0.47–2.92	0.741	1.01	0.38–2.70	0.988
IL-6 elevated vs. normal	3.09	1.39–6.86	0.006	2.88	1.29–6.42	0.010
CRP elevated vs. normal	2.12	0.79–5.72	0.137	3.16	1.20–8.27	0.019
YKL-40 elevated vs. normal	1.08	0.59–1.96	0.802	1.33	0.72–2.45	0.360

**Table 4 cancers-13-03892-t004:** Lead times with 95% confidence intervals (95% CI) calculated from first timepoint of elevated marker to radiological or clinical diagnosis of local relapse or metastatic disease.

	*n*	Median (Months)	95% CI
CEA elevated	29	7.8	5.7–9.8
CA19-9 elevated	14	10.0	6.7–13.3
IL-6 elevated	16	21.8	4.0–39.6
CRP elevated	12	10.2	5.4–15.1
YKL-40 elevated	27	53.1	27.1–79.1
CEA, CA19-9, IL-6, CRP, or YKL-40 elevated	42	27.3	17.1–37.5

**Table 5 cancers-13-03892-t005:** Time-dependent adjusted model for disease-free survival (DFS) for 2-fold increases in CEA, CA19-9, IL-6, CRP, and YKL-40, with hazard ratios (HR) and 95% confidence intervals (95% CI).

	DFS		
	HR	95% CI	*p*-Value
Adjusted *			
CEA ^1^	2.08	1.82–2.38	<0.001
CA19-9 ^2^	1.39	1.22–1.58	<0.001
IL-6 ^3^	1.48	1.24–1.76	<0.001
CRP ^4^	1.61	1.30–1.98	<0.001
YKL-40 ^5^	1.30	1.07–1.57	0.008

* Adjusted for age, sex, inflammatory disease, chemotherapy regimen, and TNM stage. These continuous variables are log2 transformed. ^1^
*n* = 132, ^2^
*n* = 111, ^3^
*n* = 143, ^4^
*n* = 146, ^5^
*n* = 144.

## Data Availability

The data collected for this study can be made available to others in de-identified form after all primary and secondary endpoints have been published, in the presence of a data transfer agreement, and if the purpose of use complies with Finnish legislation. Requests for data sharing can be made to the corresponding author, including a proposal that must be approved by the steering committee.

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
