# Peer review of "Lead Time and Prognostic Role of Serum CEA, CA19-9, IL-6, CRP, and YKL-40 after Adjuvant Chemotherapy in Colorectal Cancer"

_cancers, 2021, doi:10.3390/cancers13153892_

Round 1
Reviewer 1 Report
Authors should be more concise in the presentation of the manuscript. Some of the text and graphs can be abbreviated and condensed.
Author Response
Authors should be more concise in the presentation of the manuscript. Some of the text and graphs can be abbreviated and condensed.
We have condensed and abbreviated the results section and moved the Figure with ROC curves to supplementary material. We have also condensed the discussion somewhat.
Reviewer 2 Report
The authors assessed the prognostic impact of serum CEA, CA19-9, IL-6, CRP, and YKL-40. Elevated biomarker levels suggested impaired prognosis. In addition, longitudinal biomarker levels suggested elevated CEA levels could predict early recurrence, while elevated IL-6 and YKL-40 could suggest late recurrence. Biomarkers were measured only one time-points in most studies assessed the association between biomarkers and prognosis. The longitudinal results were unique for the study. The results of the study would attribute to patients' care after adjuvant chemotherapy for CRC. However, there are some points for improvement.
Major points
1. The blood samples were collected multiple times. I guessed serum biomarkers were measured in all samples, but it was not clearly described.
2. It was unclear that the biomarkers at which time points were used in each analysis.
3. The competing risk was considered to calculate ROC curves, while it was not considered in other survival analyses. Is there any reason to assess the competing risk only in the ROC?
4. KM curves and HRs suggested the association between biomarkers and prognosis. Only a tiny proportion of patients have elevated biomarkers except for YKL-40 (10-20%, in table 2). Relatively low AUCs might be caused by the tiny proportion. The low AUCs suggested less predictive performance of the biomarkers. On the other hand, CEA seemed to be elevated before termination or censoring (supplementary figure 6A). Is it possible that regular CEA measurement might predict a recurrence in the near future?
Minor points
1. The median age in table 1 (60.3) should be 60.
2. The resolution of Figure 4 was lacking.
3. In supplementary Figure 6, the authors showed median and IQR of biomarkers in every time-points. I recommended adding the number of patients with elevated biomarker levels.
Author Response
MAJOR POINTS
- The blood samples were collected multiple times. I guessed serum biomarkers were measured in all samples, but it was not clearly described.
Clarified: Mat&Met: "Serum biomarkers CEA and CA19-9 were analysed prospectively as part of the clinical routine, and IL-6, CRP, and YKL-40 were measured post hoc in samples collected postoperatively (baseline) before adjuvant chemotherapy, and at 4 months, 8 months (approximately 2 months after completion of adjuvant chemotherapy and 10 months from surgery), 1 year, 2 years, 3 years, 5 years, and 10 years from the date of randomisation."
Results heading 3.6: "Diagnostic accuracy for postoperative serum biomarker levels, i.e. before adjuvant treatment, and diagnosis of recurrence"
Results heading 3.7: "3.7 Lead times, i.e. interval between elevated serum biomarker and diagnosis of recurrence, during surveillance"
Results heading 3.8: Time-varying serum biomarkers during surveillance in the Cox regression model and DFS
2. It was unclear that the biomarkers at which time points were used in each analysis.
We have clarified this in the text and in the headings once for post-adjuvant, and throughout if longitudinal measurement:
Results under heading 3.1: "The CEA and CA19-9 levels were elevated in 11–14% of patients postoperatively and in 10–12% post-adjuvant i.e. at 8 months from randomisation and the inflammatory markers IL-6, CRP, and YKL-40 were elevated in 12–44% postoperatively and 8-49% post-adjuvant (Table 2). "
Results heading 3.2.: "Association between elevated biomarkers in the post-adjuvant setting, i.e. at 8 months from randomisation, and DFS or OS"
Results heading 3.3. : "Association between post-adjuvant, i.e. at 8 months after randomisation, IL-6 and DFS or OS within TNM-stages"
Results heading 3.6: "Diagnostic accuracy for postoperative serum biomarker levels, i.e. before adjuvant treatment, and diagnosis of recurrence"
Results heading 3.7: "3.7 Lead times, i.e. interval between elevated serum biomarker and diagnosis of recurrence, during surveillance"
Results heading 3.8: Time-varying serum biomarkers during surveillance in the Cox regression model and DFS. AND in text: "The time-varying biomarker Cox-regression models were adjusted for age, sex, inflammatory disease, TNM stage, and chemotherapy regimen. A 2-fold increase during surveillance in CEA-values starting from the postoperative values was linked to a 2.1-fold higher risk of relapse or death during follow-up. "
3. The competing risk was considered to calculate ROC curves, while it was not considered in other survival analyses. Is there any reason to assess the competing risk only in the ROC?
In Table 1 and the results line 246-247 we have clearly tried to show the non-CRC deaths clearly (all events after 6.3 years are non-CRC).
In Kaplan-Meier and Cox regression analyses the competing risk is taken into account as non-CRC death is an event both for DFS and OS analyses.
In the modified model we have also evaluated this as events within one year are attributable to relapse but not later events i.e. non-CRC deaths.
We did also perform RFS and DSS analyses, but as events got rarer and as cancer relapse could not be excluded as cause of death in one case with suicide (alcohol intoxication), adjuvant treatment cannot be ruled out as related to second cancer and as a predisposing factor for cardiac events, we decided to present DFS and OS as recommended outcome measures.
In lead time analysis we only looked at cancer relapse as event of interest.
4. KM curves and HRs suggested the association between biomarkers and prognosis. Only a tiny proportion of patients have elevated biomarkers except for YKL-40 (10-20%, in table 2). Relatively low AUCs might be caused by the tiny proportion. The low AUCs suggested less predictive performance of the biomarkers. On the other hand, CEA seemed to be elevated before termination or censoring (supplementary figure 6A). Is it possible that regular CEA measurement might predict a recurrence in the near future?
We do agree that only a small proportion (8-17%) has elevated markers other than YKL-40 postoperatively and post-adjuvant and this is depicted in AUCs. YKL-40 is an inflammatory marker and thus elevated in several other conditions as well, which we tried to take into account by adjusting for inflammatory diseases in our COX analyses.
We tried to show these tiny proportions clearly in the text and added in the proportions for patients with relapse during follow-up to clarify this: "The CEA and CA19-9 levels were elevated in 11–14% of patients postoperatively and in 10–12% post-adjuvant and the inflammatory markers IL-6, CRP, and YKL-40 were elevated in 12–44% postoperatively and 8-49% post-adjuvant (Table 2). These elevated post-adjuvant proportions were 11-57% in patients with relapse during follow-up. "
And added in Supplementary figure 1 as well to show the marker profiles in follow-up: "The medians and ranges are summarised in Table 2 and Supplementary Figure 1 shows changes, medians and percentiles of all five biomarkers during surveillance."
In the post-adjuvant all cases with relapses (n=12) were excluded from further analyses and thus we have a plateau of 8+ months for all DFS and OS curves presented. Clearly all our analyses show that "that regular CEA measurement might predict a recurrence in the near future". We have added this to discussion "Based on the modified model is seems like CEA measurement would predict a recurrence only in the near future."
We agree that the AUCs are only moderate as these markers are not elevated in all cases, for example CEA is elevated in only 60-80% of cases and additionally varies with site of metastases for example more often elevated for liver than for lung mets etc. Ca19-9 is even more rarely elevated. But interestingly IL-6 and CRP showed better long-term AUCs than the traditional markers CEA and Ca19-9. PPV is high for all these markers but not depicted in AUC (see Hermunen et al Acta Oncol 2020;59(12):1416-1423) and shows the value of these markers in clinical practice. The AUCs are also diluted by non-CRC deaths as taken into account as competing events.
There is a built in bias as PO acted on elevated CEA and CA19-9 measured prospectively, but could not react on elevated IL-6 or YKL-40 as they were measured retrospectively. This means that PO did a CT and colonoscopy when elevated biomarkers were found and thus shortened the lead time for these markers. We have tried to be honest with this limitation in discussion.
Minor points
1. The median age in table 1 (60.3) should be 60.
We have used one decimal for age (60.3 adn 60.4) as upper limit for inclusion was 75 years and otherwise we would have included 76 year olds in the study, thus we prefer to keep the age with one decimal.
2. The resolution of Figure 4 was lacking.
Figure 4 has been moved to Supplement as we cannot get the resolution satisfactory with reasonable size of the file, even with adobe illustrator attempts.
3. In supplementary Figure 6, the authors showed median and IQR of biomarkers in every time-points. I recommended adding the number of patients with elevated biomarker levels.
Number of elevated patients at the postoperative and post-adjuvant timepoints are shown in Table 1. As visualised in the Supplementary Figure 6 (now Supplementary Figure 1) the markers vary very much during surveillance as treatment for metastases (metastasectomy and systemic treatment) alters the levels and thus this is not just a measure for the marker and outcome of interest - thus we preferably would not add in these numbers. If still wished we will add them.
Reviewer 3 Report
In line 350~351, you describe that CEA combining one or more biomarkers can extended the lead time 3-fold. But there is no model of combination in the article. Can you show the model or formula and tell us how many biomarkers can get the result?
Author Response
In line 350~351, you describe that CEA combining one or more biomarkers can extended the lead time 3-fold. But there is no model of combination in the article. Can you show the model or formula and tell us how many biomarkers can get the result?
"CEA had a shortest lead time and by combining one or more biomarkers, lead time was extended nearly 3-fold."
We made a separate section in results for this and clarified the model in statistics and results.
Statistics: "The lead time was defined as the time from elevated biomarker value (above cut off) to diagnosis of relapse, first separately for all five biomarkers and then combined as any of the five biomarkers elevated. "
"3.7 Lead times, i.e. interval between elevated serum biomarker and diagnosis of recurrence, during surveillance.
The lead time, i.e., from first timepoint of elevated serum CEA (measured at predefined intervals from postoperative to 10 years) to the radiological or clinical diagnosis of a local relapse or distant metastasis, was 7.8 months (Table 4). The lead time was 10.0 months for CA19-9, 21.8 months for IL-6, 10.2 months for CRP and 53.1 months for YKL-40 (Table 4). The lead time for one or more of the five biomarkers elevated was 27.3 months. "
The lead time from first timepoint of elevation of any (one or more) of the five biomarkers elevated to relapse was 27.3 months (Table 4).
Round 2
Reviewer 2 Report
The authors clearly responded to all the comments from the reviewer. I have no further comments.